# Knowing the learning strategy is not enough to use it: Example in reading strategies for Japanese undergraduates

**Tsuyoshi Yamaguchi** *

Liberal Arts and Sciences, Nippon Institute of Technology, Minamisaitama-gun, Saitama Pref., Japan

* yamaguchi.tsuyoshi@nit.ac.jp

## Abstract

Learning strategies are an important component of self-regulated learning. Learners are expected to use multiple strategies appropriately. This study focused on metacognitive knowledge in the use of learning strategies and attempted to clarify the hierarchical nature of multiple knowledge. Furthermore, the study provided suggestions that could lead to further efficient acquisition of learning strategies. Responses were obtained from 184 Japanese university students regarding the degree of strategy use, knowledge regarding strategy, and perceived benefit and cost of 28 reading strategies. Results of the hierarchical Bayesian modeling showed that strategy use was influenced by knowledge regarding strategy and perceived benefit and cost. Furthermore, the effects of perceived benefit and cost were lower in the absence of knowledge regarding strategy. This implies that to use a learning strategy, the learner must first be aware of it and the degree to which it is used (apart from its theoretical usefulness) is determined by subjective benefit and cost. Therefore, in classroom situations, it is desirable to explicitly teach not only the course content but also strategies appropriate for learning the content. Dependence of the effects of perceived benefit and cost of strategy use on the presence or absence of knowledge regarding strategy suggests a hierarchy of metacognitive knowledge regarding usage of learning strategies.

## Introduction

Research findings to date suggest that self-regulated learning (SRL) is an ideal form of learning [1, 2]. SRL is a proactive learning activity in which the learner assesses his or her own learning situation and adjusts both the learning method and motivation as needed. It is well recognized that the more SRL that are utilized, the better understanding of the learning content and the better academic performance [2]. Furthermore, it consists of various elements, including the appropriate use of various learning strategies and metacognition [3]. Panadero summarized the characteristics of several models of SRL and showed that learning strategies that directly and indirectly affected the acquisition of academic content were assumed in all models [3]. In Boekaerts's models [4, 5], SRL required learners to be flexible and adaptive to the task. Hence, learners were required to select an appropriate strategy for the task under the assumption that they used multiple learning strategies.

**Data Availability Statement:** All data and analysis files are available from the Open Science Framework (OSF) project (https://osf.io/wr5bs/).

**Funding:** This work was supported by the JSPS [https://www.jsps.go.jp/] KAKENHI (Grant

Numbers: JP13J04514[https://kaken.nii.ac.jp/grant/KAKENHI-PROJECT-13J04514/] & JP21K13695[https://kaken.nii.ac.jp/grant/KAKENHI-PROJECT-21K13695/]) and Special Grant (2019) by Nippon Institute of Technology [https://www.nit.ac.jp/english/]. The funders had no role in the study design, data collection and analysis, decision to publish, or manuscript preparation.

**Competing interests:** The authors have declared that no competing interests exist.

Yamaguchi referred to the assumption that one learner used multiple strategies to examine the intra-individual variance of learning strategies [6]. Examination of this assumption attempted to capture the process of an individual's use of multiple strategies, rather than compare the individual differences for a given strategy [7, 8]. Therefore, he contended that the factors that promoted and inhibited the use of learning strategies should be identified as the knowledge regarding strategy and perceived benefit and cost of the learning strategies themselves [9]. Although SRL has been focused on from an individual differences perspective, focusing on motivational variables such as "what kind of learner" [2], it is also important to approach SRL from an intra-individual perspective, such as "how each learner uses the strategy".

Schraw and Moshman also proposed three metacognitive awareness conditions as *knowledge of cognition* [9]: knowledge of the thing itself (declarative knowledge), how to handle it (procedural knowledge), and the conditions (conditional knowledge). They reviewed metacognitive theory prior to the publication of this literature and proposed two components of metacognition: not only "knowledge of cognition" as described above, but also "regulation of cognition" such as planning, monitoring, and evaluation. To date, their proposed model has been widely accepted, with metacognition proven to mediate achievement goals and facilitate mathematical modelling in mathematics education [10], and a list of metacognitive teaching practices for instructors to implement in biology education [11]. Furthermore, metacognitive knowledge has emerged as a key individual metacognitive trait in self-control studies. [12]. Murayama used their knowledge of cognition to model [9] a process to use a learning strategy hierarchically [13]. The first stage was "knowledge regarding strategy," which was declarative knowledge of the strategy itself. The second stage was "procedural knowledge," which was the (automatic) mastery of the strategy. The third stage was "perceived benefit and cost", which was the subjective perception of the learning effect and action cost of using the strategy. The fourth (final) stage was "conditional knowledge," which served to identify the third stage in further detail. Both the second and first stages were necessary conditions for using a strategy, especially the first stage. Furthermore, the third and fourth stages determined the extent to which a strategy was used after this condition was passed. However, the hierarchy of metacognitive knowledge that lead up to the use of such learning strategies has not been sufficiently investigated.

This study adopted a model of intra-individual variance of learning strategies and examined the effects of knowledge regarding strategy (*Knowledge*) and perceived benefit/cost (*Benefit/Cost*) on strategy use (*Used*) in a stepwise manner, which included an examination of interactions. First, the explanation of *Used* using *Knowledge* alone (Model 1) was examined. Subsequently, the study examined whether the explanation improved when *Benefit/Cost* was added (Model 2). In addition, whether the influence of *Benefit/Cost* was strengthened by the strategy with knowledge as a precondition (Models 3 and 4) was also investigated. This was the first study to examine the hierarchical structure of the learning strategy use process. Hence, the author decided to simplify the models to be examined. Specifically, a survey approach that allowed for a larger sample size and shorter measurement time was adopted. Furthermore, this study did not consider procedural knowledge, which was difficult to measure with this approach. Independent examination was necessary owing to the limitations of the approach and different characteristics of procedural knowledge from the perspective of long-term memory from other types of knowledge. Conditional knowledge was a further detailed classification of *Benefit/Cost* in the previous stage. However, since the hierarchy of the previous stage was unclear, this study decided not to consider it to avoid complexity. In addition, college students, who should already be able to use multiple strategies, were asked to respond to many strategies. This limited the study to the strategy of reading and comprehending explanatory texts to create

a measurement that could be assess students across various majors. Despite these limitations, the appropriate use of multiple strategies was important for the realization of SRL. Furthermore, the step-by-step process of learning strategy use that this study examined could serve as a guideline for the strategy instruction that educators should provide.

The purpose of this study is to illustrate the simplified results of the process leading up to the use of a learning strategy, in particular the hierarchical nature of the process due to metacognitive knowledge. If a learning strategy is used progressively, then (a) the learner must first comprehend the strategy to use it, and (b) after comprehending it, the learner determines whether or not to use it based on subjective benefit and cost. These predictions are supported when (a') the main effect of *Knowledge* on *Used* is observed and (b') the effects of *Benefit* and *Cost* on *Used* are observed in strategies with *Knowledge*.

## Methods

### Ethical considerations

This study was approved by the Human Research Ethics Committee of the Ethical Review Board of Major Psychology of Hosei University (date of approval: June 20th, 2012). Ethical considerations were explained to the participants both verbally and in writing before the survey was initiated. Participants were asked to sign their affiliation and name when participating in this study. After collecting the survey forms including the signatures, the signed forms were separated from the research data questionnaires and stored separately. Individuals could not be identified from the data.

### Procedure and participants

Students from three universities in Tokyo, Japan, voluntarily participated in the survey, which was conducted during a single offline lecture of one course at each university in 2012. (With the exception of one university, surveys were also conducted the week before the survey for this study was conducted, but were excluded from the data and analysis because they were not suitable for the research purposes of this study.)

In total, 189 students agreed to participate. Of these, data from 184 participants were analyzed (74 females and 109 males; $M_{age}$ = 20.27 years, $SD_{age}$ = 1.15, and Range = 18–25 years). Of the participants, five were excluded from the dataset for the following reasons: participants whose age was more than two standard scores away from the others ($n = 2$), responses ended in the middle of the questionnaire ($n = 2$), and did not respond to all the variables ($n = 1$). To examine the hierarchy of metacognitive knowledge of reading strategy use in college students' descriptive essays, age was used as a criterion in this study to exclude graduate students and experienced working adults, who are likely to have more experience reading expository texts than college students. Regarding the missing data criteria, the rationale is that abandoning of a response in the middle of the survey is not random missing data, and it may cause bias in the results. [14]. As described below, there were 28 reading strategies, and participants were asked to respond to four variables (*Used*, *Knowledge*, *Benefit*, and *Cost*) per item, so the maximum response per participant was 112. Of the 184 participants who were not excluded by the above criteria, 22 had at least one missing measurement (22 / 184 = 12%), and even the participant with the most missing measurements among these 22 had a missing number of 12 (12 / 112 = 11%).

### Measures

Participants were asked to respond to one strategy item in the following order: actually used (Used), knowledge regarding strategy (Knowledge), perceived benefit (Benefit), and perceived

cost (Cost). Participants responded to the degree of Used, Benefit, and Cost on a 6-point Likert scale that ranged from 1 (not at all true) to 6 (extremely true). Furthermore, they responded to the knowledge on a binary scale (I (A) was aware of or (B) was not aware of the existence of this method).

Variance in participants' responses was required to calculate the within-person correlations. Participants completed a questionnaire that consisted of four items in each of the seven categories that reflected reading strategies (i.e., clarifying, grasping the points, memorizing, noticing the text structure, utilizing the knowledge, monitoring, and control). Therefore, there were 28 items in total. This questionnaire's categories and items were referenced in the Reading Strategies Questionnaire developed by Inuzuka [15] based on his findings on the process of comprehension of expository texts [16–18]. Table 1 shows the mean, standard deviation, and number of missing items for each of the 28 items and four variables. The description of each item, categories assumed, and instructional text provided by the participants are in the Supporting information. (Other questionnaire about achievement goals were also asked. However, they were not addressed because they did not meet the objectives of this study).

**Table 1. Participant means (*M*) and standard deviation (*SD*) of each assessment item and the number of people with and without prior strategy knowledge.**

| Item ID | Used | | Knowledge[a] | | Benefit | | Cost | |
|---|---|---|---|---|---|---|---|---|
| | *M* | *SD* | Knew | Did not | *M* | *SD* | *M* | *SD* |
| 1 | 3.03 | 1.39 | 101 | 82 | 4.26 | 1.11 | 3.88 | 1.35 |
| 2 | 4.40 | 1.13 | 175 | 9 | 4.39 | 1.08 | 3.71 | 1.43 |
| 3 | 3.28 | 1.53 | 147 | 37 | 4.28 | 1.18 | 3.86 | 1.55 |
| 4 | 4.05 | 1.24 | 168 | 15 | 4.50 | 1.10 | 3.05 | 1.22 |
| 5 | 3.04 | 1.33 | 135 | 47 | 3.07 | 1.32 | 3.17 | 1.31 |
| 6 | 3.84 | 1.31 | 163 | 21 | 4.46 | 1.01 | 3.73 | 1.31 |
| 7 | 4.11 | 1.28 | 173 | 11 | 4.61 | 1.01 | 3.84 | 1.32 |
| 8 | 2.96 | 1.48 | 136 | 48 | 3.86 | 1.30 | 4.16 | 1.43 |
| 9 | 4.22 | 1.12 | 168 | 16 | 4.42 | 0.99 | 3.14 | 1.26 |
| 10 | 3.92 | 1.17 | 167 | 15 | 4.34 | 0.98 | 3.52 | 1.28 |
| 11 | 4.70 | 1.38 | 179 | 5 | 4.65 | 1.21 | 2.77 | 1.37 |
| 12 | 4.55 | 1.17 | 173 | 11 | 4.41 | 1.09 | 2.96 | 1.31 |
| 13 | 3.53 | 1.31 | 149 | 35 | 4.37 | 1.04 | 3.79 | 1.29 |
| 14 | 4.73 | 0.83 | 179 | 5 | 4.84 | 0.83 | 3.10 | 1.19 |
| 15 | 2.38 | 1.26 | 93 | 90 | 2.56 | 1.19 | 3.85 | 1.47 |
| 16 | 3.54 | 1.56 | 112 | 72 | 4.06 | 1.14 | 3.02 | 1.37 |
| 17 | 2.40 | 1.22 | 110 | 74 | 2.15 | 1.13 | 3.89 | 1.60 |
| 18 | 3.42 | 1.36 | 120 | 62 | 4.21 | 1.21 | 3.31 | 1.33 |
| 19 | 3.22 | 1.32 | 123 | 59 | 3.98 | 1.15 | 3.43 | 1.32 |
| 20 | 4.34 | 1.14 | 165 | 17 | 4.36 | 0.99 | 3.59 | 1.35 |
| 21 | 3.75 | 1.32 | 134 | 50 | 4.20 | 1.08 | 3.28 | 1.36 |
| 22 | 2.45 | 1.36 | 143 | 40 | 4.46 | 1.21 | 4.82 | 1.34 |
| 23 | 4.45 | 1.13 | 169 | 12 | 4.50 | 1.00 | 3.95 | 1.36 |
| 24 | 3.86 | 1.43 | 167 | 15 | 4.13 | 1.21 | 4.17 | 1.38 |
| 25 | 3.75 | 1.30 | 140 | 40 | 4.23 | 1.09 | 3.69 | 1.40 |
| 26 | 4.28 | 1.13 | 165 | 15 | 4.53 | 0.99 | 3.48 | 1.42 |
| 27 | 3.98 | 1.25 | 155 | 26 | 4.49 | 1.01 | 3.37 | 1.38 |
| 28 | 4.43 | 1.09 | 159 | 22 | 4.73 | 0.92 | 2.94 | 1.27 |

*Note. Used, Benefit*, and *Cost* were all scored in the range of 1 to 6.

[a]The number of respondents is shown for each strategy.

## Data analysis

In total, four models (five if the model with no independent variable to calculate the intraclass correlation coefficient was included) in which the independent variable and its interaction terms were put in order were compared regarding goodness of fit. For each model, random and fixed effects were considered, [19], and a Markov chain Monte Carlo (MCMC) method was employed to estimate appropriate parameters [20]. Mplus ver. 8.3 was used to perform such hierarchical Bayesian modelling [21]. Furthermore, the results that showed the most reasonable fit with the data were referenced. The models were shown via the following equation:

$$
\begin{aligned}
Used_{ij} \quad = \ & \beta_{0ij} \\
& + \beta_{1ij}(Knowledge)_{ij} + \beta_{2ij}(Benefit)_{ij} + \beta_{3ij}(Cost)_{ij} \\
& + \beta_{4ij}(Knowledge)_{ij}(Benefit)_{ij} + \beta_{5ij}(Knowledge)_{ij}(Cost)_{ij} \\
& + \beta_{6ij}(Benefit)_{ij}(Cost)_{ij} + \beta_{7ij}(Knowledge)_{ij}(Benefit)_{ij}(Cost)_{ij} \\
& + r_{ij}
\end{aligned}
\tag{1}
$$

where the dependent variable, $(Used)_{ij}$, represented the amount of strategy use for item $i$ and participant $j$. Model 0 determined the within-individual (item-to-item) and between-individual variances in the degree of use of the strategy by means of the parameter $\beta_{0ij}$. These were referred to in the equations below. Model 1 added a $(Knowledge)_{ij}$ term to Model 0, and Model 2 added a $(Benefit)_{ij}$ and $(Cost)_{ij}$ term to Model 1. Each parameter $\beta_{1ij}$, $\beta_{2ij}$, and $\beta_{3ij}$ represented a linear effect on the dependent variable and inter- and intra-individual variation of that effect. Model 3 showed first-order interaction effects and Model 4 showed second-order interaction effects, denoted by the parameter $\beta_{4ij}$, $\beta_{5ij}$, $\beta_{6ij}$, and $\beta_{7ij}$. A common residual $r_{ij}$ was assumed for all the models. Furthermore, each parameter $\beta$ was decomposed into three parameters: fixed effects $\gamma$, between-participant variation $\mu$, and between-item variation $v$ [6, 19], as follows:

$$
\beta_{1ij} = \gamma_{10} + \mu_{1j} + v_{1i}
\tag{2}
$$

$$
\beta_{2ij} = \gamma_{20} + \mu_{2j} + v_{2i}
\tag{3}
$$

$$
\beta_{3ij} = \gamma_{30} + \mu_{3j} + v_{3i}
\tag{4}
$$

$$
\beta_{4ij} = \gamma_{40} + \mu_{4j} + v_{4i}
\tag{5}
$$

$$
\beta_{5ij} = \gamma_{50} + \mu_{5j} + v_{5i}
\tag{6}
$$

$$
\beta_{6ij} = \gamma_{60} + \mu_{6j} + v_{6i}
\tag{7}
$$

$$
\beta_{7ij} = \gamma_{70} + \mu_{7j} + v_{7i}
\tag{8}
$$

Subsequently, the variances, $\tau$ and $\omega$, of the parameters $\mu$ and $\nu$, indicated the individual and within-individual random effects.

$$Var(\mu_{1j}) = \tau_{11},$$
$$Var(\mu_{2j}) = \tau_{22},$$
$$Var(\mu_{3j}) = \tau_{33},$$
$$Var(\mu_{4j}) = \tau_{44}, \tag{9}$$
$$Var(\mu_{5j}) = \tau_{55},$$
$$Var(\mu_{6j}) = \tau_{66},$$
$$Var(\mu_{7j}) = \tau_{77}.$$

$$Var(\nu_{1i}) = \omega_{11},$$
$$Var(\nu_{2i}) = \omega_{22},$$
$$Var(\nu_{3i}) = \omega_{33},$$
$$Var(\nu_{4i}) = \omega_{44}, \tag{10}$$
$$Var(\nu_{5i}) = \omega_{55},$$
$$Var(\nu_{6i}) = \omega_{66},$$
$$Var(\nu_{7i}) = \omega_{77}.$$

Although this study assumed a hierarchical model that allowed for random effects of participants and items, the interpretation of the effects of the variables, such as knowledge regarding strategy on usage of strategy, referred primarily to fixed effects. This was since this study was interested in the effects of the variables, such as strategy knowledge on strategy use. Furthermore, the random effects on strategy use were still poorly known. In addition, since it was difficult to assume normality for such a large number of parameters, MCMC method was used to estimate the parameters [20]. The estimate was the median of the posterior distribution, and this study referred to the 95% credible interval for significance regarding statistical hypothesis testing by Mplus ver. 8.3 [21]. The deviance information criterion (DIC) was used to compare the models. It penalized complex models and had the same interpretation as the other information criterion (i.e., the model with the smaller value was adopted). It was used when the MCMC methods were used to obtain estimates.

To avoid multicollinearity between the terms of each independent variable and interaction terms and isolate the between-individual and within-individual variances [22], the binary variable *Knowledge* was subjected to effects coding ("Knew" coded 0.5 and "Did not know" coded -0.5). Meanwhile, *Benefit* and *Cost* were subjected to a procedure in which the mean value per participant was subtracted from each item's value.

## Results

Table 2 shows the mean, standard deviation, intra-class correlation (ICC; bolded in Table 2), and within-person correlation coefficient. First, the ICC coefficients were low ($.12 < ICCs < .27$), which indicated that there was a large intra-individual (inter-item) variation in all the

**Table 2. Descriptive statistics and within-person and intraclass correlation coefficients.**

|  | Variable | $M$ | $SD$ | Missing | 1 | 2 | 3 | 4 |
|---|---|---|---|---|---|---|---|---|
| 1 | *Used* | 3.74 | 0.57 | 7 | **.13** | | | |
| 2 | *Knowledge* | 148.86 | 33.96 | 33 | .49 | **.13** | | |
| 3 | *Benefit* | 4.18 | 0.52 | 8 | .58 | .33 | **.15** | |
| 4 | *Cost* | 3.55 | 0.77 | 10 | -.38 | -.11 | -.16 | **.26** |

*Note. M*, *SD*, and Missing are the mean, standard deviation, and total number of missing values, respectively. Diagonals in the correlation matrix indicate the intraclass correlation coefficient.

variables. Second, there was a positive intra-individual correlation between *Used* and *Knowledge* and *Benefit* ($r_{Used-Knowledge}$ = .49, $r_{Used-Benefit}$ = .58). Furthermore, there was a negative correlation between *Used* and *Cost* ($r_{Used-Cost}$ = −.38). This trend was similar to that in previous studies that examined intra-individual correlations in learning strategy research [6–8].

Since the assumption of intra-individual variance and relationship between the intra-individual correlations were confirmed to be similar to those in previous studies, the fit between the model and data was confirmed. Table 3 shows the DIC, number of parameters, and whether 95% credible intervals for the parameter estimates represented the fixed effects included (*n.s.*) or excluded (*sig.*) zero. The DIC value was lowest for Model 4 ($DIC_{Model4}$ = 12655.37), which was injected up to a second-order interaction term. Hence, Model 4 should be adopted. However, the 95% credible interval for the fixed effect of the crucial second-order interaction term included 0. Therefore, the results of Model 3, which included up to a first-order interaction term and whose DIC values were not far from those of Model 4 ($DIC_{Model3}$ = 12659.52), were used as reference.

Table 4 shows the results of Model 3, which included up to a first-order interaction term. The fixed effects revealed the following trends: a 1.38 increase in *Used* scores for the strategy with *Knowledge* compared to without *Knowledge* ($\gamma_{10}$ = 1.38 [1.25 − 1.52]), 0.36 increase in *Used* scores for a 1-point increase in *Benefit* scores ($\gamma_{20}$ = 0.36 [0.31 − 0.41]), and 0.22 decrease in *Used* scores for a 1-point increase in *Cost* scores ($\gamma_{30}$ = −0.22[−0.27 − −0.16]). Furthermore, the effects of *Benefit* and *Cost* on *Used* varied based on the presence or absence of *Knowledge* ($\gamma_{40}$ = 0.35[0.27 − 0.43], $\gamma_{50}$ = −0.12[−0.20 − −0.03]). There was no interaction effect between *Benefit* and *Cost* ($\gamma_{60}$ = −0.03[−0.06 − 0.01]). Regarding the random effects, there was variance in *Used* scores across the participants ($\tau_{00}$ = 0.20[0.16 − 0.26]) and items ($\omega_{00}$ = 0.10[0.06 − 0.19]) as well as an effect of participant variance on the effect of *Knowledge* on *Used* ($\tau_{11}$ = 0.28[0.20 − 0.40]). Other random effects were nearly 0.

**Table 3. Deviance information criterion (DIC) of each model and fixed effects that were significant as a result of the analysis of each model.**

| | Model fit information | | | Whether fixed effects are significant | | | | | | |
|---|---|---|---|---|---|---|---|---|---|---|
| Models | DIC | pD | Free | Knowledge | Benefit | Cost | KB | KC | BC | KBC |
| Model 1 | 14860.13 | 284.39 | 7 | sig. | - | - | - | - | - | - |
| Model 2 | 12873.29 | 488.22 | 13 | sig. | sig. | sig. | - | - | - | - |
| Model 3 | 12659.52 | 587.45 | 22 | sig. | sig. | sig. | sig. | sig. | n.s. | - |
| Model 4 | 12655.37 | 604.13 | 25 | sig. | sig. | sig. | sig. | sig. | sig. | n.s. |

*Note. pD* and *Free* are number of valid and free parameters, respectively. *KB*, *KC*, *BC*, and *KBC* denote the combination of interaction terms and are the first letter of each variable. *sig.* is the term that did not contain 0 in the 95% confidence interval and *n.s.* is the term that contained 0.

**Table 4. Results of the hierarchical Bayesian analysis for Model 3 which includes first-order-interactions.**

|  | Variable | Estimation | | |
|---|---|---|---|---|
|  |  | *Lower* | *Median* | *Upper* |
| Coefficient (Fixed effects) | | | | |
| $\gamma_{00}$ | *(Intercept)* | 3.10 | 3.24 | 3.38 |
| $\gamma_{10}$ | *Knowledge* | 1.25 | 1.38 | 1.52 |
| $\gamma_{20}$ | *Benefit* | 0.31 | 0.36 | 0.40 |
| $\gamma_{30}$ | *Cost* | -0.27 | -0.22 | -0.16 |
| $\gamma_{40}$ | *KB* | 0.27 | 0.35 | 0.43 |
| $\gamma_{50}$ | *KC* | -0.20 | -0.12 | -0.03 |
| $\gamma_{60}$ | *BC* | -0.06 | -0.03 | 0.01 |
| $r_{ij}$ | *(residual)* | 0.60 | 0.62 | 0.65 |
| Variances by Participants (Random effects) | | | | |
| $\tau_{00}$ | *(Intercept)* | 0.16 | 0.20 | 0.26 |
| $\tau_{11}$ | *Knowledge* | 0.20 | 0.28 | 0.40 |
| $\tau_{22}$ | *Benefit* | 0.01 | 0.02 | 0.03 |
| $\tau_{33}$ | *Cost* | 0.01 | 0.01 | 0.02 |
| $\tau_{44}$ | *KB* | 0.00 | 0.02 | 0.05 |
| $\tau_{55}$ | *KC* | 0.01 | 0.03 | 0.06 |
| $\tau_{66}$ | *BC* | 0.00 | 0.00 | 0.01 |
| Variances by Items (Random effects) | | | | |
| $\omega_{00}$ | *(Intercept)* | 0.06 | 0.10 | 0.19 |
| $\omega_{11}$ | *Knowledge* | 0.00 | 0.02 | 0.07 |
| $\omega_{22}$ | *Benefit* | 0.00 | 0.00 | 0.01 |
| $\omega_{33}$ | *Cost* | 0.00 | 0.01 | 0.02 |
| $\omega_{44}$ | *KB* | 0.00 | 0.01 | 0.04 |
| $\omega_{55}$ | *KC* | 0.00 | 0.01 | 0.03 |
| $\omega_{66}$ | *BC* | 0.00 | 0.00 | 0.01 |

*Note. KB*, *KC*, *BC*, and *KBC* denote the combination of interaction terms and are the first letters of each variable.

Since there were interactions between *Knowledge* and *Benefit* and *Knowledge* and *Cost* for *Used*, the simple slope of *Benefit* and *Cost* to *Used* was examined in the presence and absence of *Knowledge*, respectively. As a rough trend, the influence of *Benefit* and *Cost* on *Used* was strengthened when *Knowledge* was present. Specifically, a 1-point increase in *Benefit/Cost* score for the strategy with *Knowledge* led to a 0.53-point increase and 0.27 decrease in *Used* score ($\gamma_{20|Knew} = 0.53[0.48 - 0.58]$, $\gamma_{30|Knew} = -0.27[-0.33 - -0.22]$). In addition, a 0.18-point increase and 0.16 decrease was observed for the strategy without *Knowledge* ($\gamma_{20|Didnotknow} = 0.18[0.11 - 0.26]$, $\gamma_{30|Didnotknow} = -0.16[-0, 24 - -0.08]$).

## Discussion

This study examined the hierarchy of metacognitive knowledge in the use of learning strategies [9, 13]. Within-person correlations of undergraduates' reading strategies showed that the knowledge regarding strategy and the perceived benefit and cost influenced the usage of a strategy. Furthermore, the influence of perceived benefit and cost varied based on whether knowledge regarding the strategy was present. In addition, there was no interaction between perceived benefit and cost.

Several studies that focused on intra-individual variance in learning strategy use also revealed that knowledge regarding a strategy and the perceived benefit and cost affected

strategy use [6–8]. This study confirmed these results and revealed the possibility of a hierarchy from the interaction. Simply, without knowing the thing itself, it would be impossible to evaluate its benefits and costs. Although this may seem obvious, it suggested the necessity of teaching both the subject matter and strategy in education. Furthermore, knowing a strategy does not necessarily mean using it. It depends on the learner's subjectivity, such as the perceived benefit and cost. If the learner's subjective view was consistent with the theoretical learning effects, it may be a situation in which SRL was being appropriately implemented. However, unfortunately, even university students reportedly did not always use theoretically effective strategies [23, 24]. Garner suggested that strategies with low costs and learning effects tended to be used as they yielded reasonable scores [25]. Furthermore, experimental approaches also revealed that when outcomes remained the same, the strategy with the lowest cost was considered [26, 27]. Therefore, in the practice of strategy instruction, it was necessary to provide opportunities to use strategies to lower their psychological cost and for learners to feel their effectiveness.

Several recommendations can be made based on the study's findings. Although the importance of teaching a knowledge regarding strategy has already been stated [11], it is possible that learners may be referring to subjective benefit and/or cost in using strategies. Based on these considerations, it is suggested that there is a need for a phase in which students are not only taught the learning strategies throughout the class, but are also provided with assignments in which they use the strategies they are aiming to acquire, and are given feedback on how their scores have increased through their use.

Although there was no interaction effect on strategy use when it came to perceived benefit and cost, procedural knowledge, which could not be included in the study variables, may have been involved. Basic experiments on reinforcement learning revealed a trade-off relationship, where benefits decreased as costs increased [27]. In this study, the reading strategy was not immediately effective and difficult to use in certain cases. This may be explained regarding acquisition of procedural knowledge. Procedural knowledge was the knowledge required to master a strategy. Furthermore, the findings of procedural memory required repeated experience with the use of a strategy and automation of its physical and cognitive works [28]. While knowledge regarding strategy, perceived benefit/cost, and conditional knowledge were acquired, declarative, procedural memory was non-declarative and differed based on the brain regions explicitly involved [29]. If a reading strategy was known linguistically, its effectiveness and cost could be evaluated. However, if procedural knowledge was not acquired, its use could have an increased cost. The process of acquiring procedural knowledge and reducing costs by automating the use of strategies should be examined experimentally and verified by an intervention in the future.

This study's conclusions are based on a self-report survey using a questionnaire. Similarly, it has been demonstrated that strategies with knowledge regarding strategy are used [6], and that strategies with higher perceived benefit/cost are used more often/less often [6–8]. This study suggests a hierarchy of metacognitive knowledge through interaction effects. Thus, there appears to be a hierarchy of metacognitive knowledge in the use of learning strategies in the learner's awareness. However, substantiating such an assumption requires controlled experimental and empirical research findings, such as a graded intervention experiment that reflects the hierarchy of assumed metacognitive knowledge.

This study has certain limitations, including the fact that it does not assess procedural knowledge, does not examine the cost-benefit trade-off, and is not an experimentally controlled study. In addition, since this was a cross-sectional study, it will be necessary to consider within-individual variance measured at multiple time points for the same variables [30]. Despite these issues, this study was the first to demonstrate the possibility of hierarchical

metacognitive knowledge of learning strategy use, even partially. These findings will lead to further refinement of the SRL model and further effective instructions.

## Supporting information

**S1 File. Explanatory text regarding responses to participants.**
(ZIP)

**S2 File. Scaling of participants' responses to reading strategy item.**
(ZIP)

**S3 File. Reading strategy items.**
(ZIP)

## Acknowledgments

This paper is a reanalysis and reorganization of a poster presented at the 15th Biennial EARLI Conference for Research on Learning and Instruction, part of a master's thesis, and part of a doctoral thesis. The survey was conducted while the author was a student at Hosei University. The author would like to thank Prof. Dr. Tetsuya Fujita and Prof. Dr. Kou Murayama. In addition, the author would also like to thank the teachers who cooperated in the survey and colleagues for their help in submitting the paper.

## Author Contributions

**Conceptualization:** Tsuyoshi Yamaguchi.

**Data curation:** Tsuyoshi Yamaguchi.

**Formal analysis:** Tsuyoshi Yamaguchi.

**Funding acquisition:** Tsuyoshi Yamaguchi.

**Investigation:** Tsuyoshi Yamaguchi.

**Methodology:** Tsuyoshi Yamaguchi.

**Project administration:** Tsuyoshi Yamaguchi.

**Validation:** Tsuyoshi Yamaguchi.

**Writing – original draft:** Tsuyoshi Yamaguchi.

**Writing – review & editing:** Tsuyoshi Yamaguchi.

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
