## [Decision Letter · Decision Letter 0]

16 Aug 2023

PONE-D-23-13187Knowing the learning strategy is not enough to use it: Example in reading strategies for Japanese undergraduatesPLOS ONE

Dear Dr. Yamaguchi,

Thank you for submitting your manuscript to PLOS ONE. After careful consideration, we feel that it has merit but does not fully meet PLOS ONE’s publication criteria as it currently stands. Therefore, we invite you to submit a revised version of the manuscript that addresses the points raised during the review process.

In the “Procedure and Participants” section, it is not clear how the authors administered the surveys. Online, offline, during the lecture, etc. The following text is also not clear/incomprehensible.

Online, 76 authors stated that “Of the participants, five were excluded from the dataset for the following reasons: participants whose age was more than two standard scores away from the others (n = 2), responses ended in the middle of the questionnaire (n = 2)”. The question obviously arises, why did the authors exclude these cases? What effect could have been on the results If they were included?

The text “Of the 184 participants, 22 had at least one missing response (22 / 184 = 12%), with 12 being the most common missing value (12 / 112 = 11%).” is incomprehensible. Specifically, what does the value 12 mean?

Which software did the authors use to analyze the data or they calculated all the mathematics by themselves? There is no need to write that mathematics if authors did not devise the method by themselves and used someone else’s model. Just mentioning the name of the model suffices.

Furthermore, the manuscript used only subjective/perception-based measures. The results cannot be validated unless a practical experiment to determine the learning effects is conducted. This is especially true as the authors claim the study was the first of its kind that determined the effect of metacognitive knowledge in the use of learning strategies and attempted to clarify the hierarchical nature of multiple knowledge.

For validation, the authors could also argue in context using the literature on the importance/effectiveness/correlation of perception/subjective ratings.

We look forward to receiving your revised manuscript.

Kind regards,

Iftikhar Ahmed Khan

Academic Editor

PLOS ONE

Journal Requirements:

Reviewers' comments:

Reviewer's Responses to Questions

**Comments to the Author**

1. Is the manuscript technically sound, and do the data support the conclusions?

Reviewer #1: Yes

2. Has the statistical analysis been performed appropriately and rigorously? 

Reviewer #1: I Don't Know

3. Have the authors made all data underlying the findings in their manuscript fully available?

Reviewer #1: Yes

4. Is the manuscript presented in an intelligible fashion and written in standard English?

Reviewer #1: Yes

5. Review Comments to the Author

Reviewer #1: This paper focused on metacognitive knowledge in the use of learning strategies which are an important aspect of self-regulated learning.

The abstract sets out the background context and rationale. As a reader, it could be clearer what is a specific key result. There are some general statements which could be positioned to emphasize for the reader a key result or takeaway and finally an overall concluding statement on the impact on practice.

Research findings till date suggest that self-regulated learning (SRL) is an ideal form of 2 learning [1, 2]. Compared to? What does ideal form of learning mean?

The opening statement needs to set the context of the research and this doesn’t do that well enough in my opinion.

Line 2: Research findings till date – to date?

Line 16/17: This was since motivational variables were individual difference variables (is not a coherent statement for the reader)

The authors cite Schraw and Moshman which is the underpinning research that informs the model used in this study. The reference is 1995 and yet no commentary on this research is given. Has it been cited or used extensively? What evidence beyond the original research supports its use? Are there any limitations with this research paper?

What is the research aim or research question? This would be helpful at the end of the introduction to bring together how this research addresses a gap in the knowledge base.

Line 94: were referenced the Reading Strategies Questionnaire – “in” the Reading?

In Table 2 there are missing values, yet the procedure and participants section, mentions exclusion of participants where variables were missing?

The discussion section, introduces the overall results which do seem pretty obvious. It is good that this is reflected because declarative knowledge would be required to evaluate perceived benefit and cost and hence the choice to use such a strategy.

it suggested the necessity of teaching both the subject matter and 205 strategy in education

A section or discussion on the impact of the outcomes from this study on practice would be useful and help contextualise the findings further.

6. PLOS authors have the option to publish the peer review history of their article (what does this mean?). If published, this will include your full peer review and any attached files.

Reviewer #1: No

---

## [Author Response · Author response to Decision Letter 0]

23 Sep 2023

Dr. Iftikhar Ahmed Khan

Academic Editor

PLOS ONE

Dear Dr. Khan,

I would like to thank you for not only managing my manuscript [PONE-D-23-13187] as an Academic Editor, but also for your suggestions on how to improve it. I am grateful for your advice to make my manuscript more readable. I have addressed each of your comments and suggestions as follows.

In the “Procedure and Participants” section, it is not clear how the authors administered the surveys. Online, offline, during the lecture, etc. The following text is also not clear/incomprehensible.

RESPONSE: Thank you for pointing this out. I have added a note.

BEFORE: line 69

Students from three universities in Tokyo, Japan, voluntarily participated in the survey, which was conducted during a lecture of one course at each university in 2012.

AFTER: line 90

Students from three universities in Tokyo, Japan, voluntarily participated in the survey, which was conducted during a single offline lecture of one course at each university in 2012.

Online, 76 authors stated that “Of the participants, five were excluded from the dataset for the following reasons: participants whose age was more than two standard scores away from the others (n = 2), responses ended in the middle of the questionnaire (n = 2)”. The question obviously arises, why did the authors exclude these cases? What effect could have been on the results If they were included?

RESPONSE: I did not explain myself well enough. The reason has been added. An additional reference was added, which caused a change in the numbering of the list of references.

BEFORE: N/A

AFTER: Line 101

To examine the hierarchy of metacognitive knowledge of reading strategy use in college students' descriptive essays, age was used as a criterion in this study to exclude graduate students and experienced working adults, who are likely to have more experience reading expository texts than college students. Regarding the missing data criteria, the rationale is that abandoning of a response in the middle of the survey is not random missing data, and it may cause bias in the results. [14].

14. Enders CK. Applied missing data analysis. New York: The Guilford Press; 2010.

The text “Of the 184 participants, 22 had at least one missing response (22 / 184 = 12%), with 12 being the most common missing value (12 / 112 = 11%).” is incomprehensible. Specifically, what does the value 12 mean?

RESPONSE: I have provided the figures, but they were insufficiently explained. Further information has been added.

BEFORE: Line 79

Of the 184 participants, 22 had at least one missing response (22 / 184 = 12%), with 12 being the most common missing value (12 / 112 = 11%).

AFTER: Line 107

As described below, there were 28 reading strategies, and participants were asked to respond to four variables (Used, Knowledge, Benefit, and Cost) per item, so the maximum response per participant was 112. Of the 184 participants who were not excluded by the above criteria, 22 had at least one missing measurement (22 / 184 = 12%), and even the participant with the most missing measurements among these 22 had a missing number of 12 (12 / 112 = 11%).

Which software did the authors use to analyse the data or they calculated all the mathematics by themselves? There is no need to write that mathematics if authors did not devise the method by themselves and used someone else’s model. Just mentioning the name of the model suffices.

RESPONSE: I found the logical structure to be difficult to understand. Thank you for your important question. I described software and models in the first paragraph of "Data analysis."

BEFORE: N/A

AFTER: Line 136

For each model, random and fixed effects were considered [19], and a Markov chain Monte Carlo (MCMC) method was employed to estimate appropriate parameters [20].

Mplus ver. 8.3 was used to perform such hierarchical Bayesian modelling [21].

Furthermore, the manuscript used only subjective/perception-based measures. The results cannot be validated unless a practical experiment to determine the learning effects is conducted. This is especially true as the authors claim the study was the first of its kind that determined the effect of metacognitive knowledge in the use of learning strategies and attempted to clarify the hierarchical nature of multiple knowledge.

For validation, the authors could also argue in context using the literature on the importance/effectiveness/correlation of perception/subjective ratings.

RESPONSE: As you point out, this study relies on self-reporting and not empirical reporting. Although there is literature that reports learning effects on metacognition in a self-report format [10], I have not found any literature that warrants the hierarchy of metacognitive knowledge that I propose. As you have highlighted, this is my first attempt, but the fact that it is not an empirical study is a limitation of this study, so I have decided to describe it as follows (AFTER). Since the present research first wanted to publicize the possibility that there is a hierarchy of metacognitive knowledge in the use of learning strategies, I introduced other studies that mention the influence of metacognitive knowledge on the use of learning strategies, including the results of this study, and specified that empirical studies are needed.

BEFORE: N/A

AFTER: Line 274

This study's conclusions are based on a self-report survey using a questionnaire. Similarly, it has been demonstrated that strategies with knowledge regarding strategy are used [6], and that strategies with higher perceived benefit/cost are used more often/less often [6-8]. This study suggests a hierarchy of metacognitive knowledge through interaction effects. Thus, there appears to be a hierarchy of metacognitive knowledge in the use of learning strategies in the learner's awareness. However, substantiating such an assumption requires controlled experimental and empirical research findings, such as a graded intervention experiment that reflects the hierarchy of assumed metacognitive knowledge.

This study has certain limitations, including the fact that it does not assess procedural knowledge, does not examine the cost-benefit trade-off, and is not an experimentally controlled study. In addition, …

++++++++++++++++++++++++++++++++++++++++++++++++++++++++++++++++++++

Dear Reviewer,

I am grateful to you for your important advice on my manuscript. I hope my correction is in line with your intention. I have addressed each of your comments and suggestions as follows.

The abstract sets out the background context and rationale. As a reader, it could be clearer what is a specific key result. There are some general statements which could be positioned to emphasize for the reader a key result or takeaway and finally an overall concluding statement on the impact on practice.

RESPONSE: Thanks for the suggestion. I have added two sentences highlighting the results and recommendations for practice.

BEFORE: N/A

AFTER: Abstract

… Furthermore, the effects of perceived benefit and cost were lower in the absence of knowledge regarding strategy. This implies that to use a learning strategy, the learner must first be aware of it and the degree to which it is used (apart from its theoretical usefulness) is determined by subjective benefit and cost. Therefore, in classroom situations, it is desirable to explicitly teach not only the course content but also strategies appropriate for learning the content. Dependence of the effects of perceived benefit and cost of strategy use on the presence or absence of knowledge regarding strategy suggests a hierarchy of metacognitive knowledge regarding usage of learning strategies.

Research findings till date suggest that self-regulated learning (SRL) is an ideal form of 2 learning [1, 2]. Compared to? What does ideal form of learning mean?

The opening statement needs to set the context of the research and this doesn’t do that well enough in my opinion.

RESPONSE: Thank you for your suggestion. I have added a note about self-adjusted learning.

BEFORE: N/A

AFTER: Line 3

… an ideal form of learning [1,2]. SRL is a proactive learning activity in which the learner assesses his or her own learning situation and adjusts both the learning method and motivation as needed. It is well recognized that the more SRL that are utilized, the better understanding of the learning content and the better academic performance [2]. Furthermore, …

Line 2: Research findings till date – to date?

RESPONSE: Thank you for pointing this out. I have reflected it in the statement.

BEFORE: Line 2

Research findings till date suggest that self-regulated learning (SRL) is an ideal form of learning [1,2].

AFTER: Line 2

Research findings to date suggest that self-regulated learning (SRL) is an ideal form of learning [1,2].

Line 16/17: This was since motivational variables were individual difference variables (is not a coherent statement for the reader)

RESPONSE: It was too complicated to explain. The claim, including the preceding statement, was ambiguous and has been corrected.

BEFORE: Line 14

Therefore, Schraw and Moshman proposed that the factors that promoted and inhibited the use of learning strategies should be taken as the knowledge regarding strategy and perceived benefit and cost of the learning strategies themselves [9]. This was since motivational variables were individual difference variables, which were important factors in SRL.

AFTER: Line 18

Therefore, he contended that the factors that promoted and inhibited the use of learning strategies should be identified as the knowledge regarding strategy and perceived benefit and cost of the learning strategies themselves [9]. Although SRL has been focused on from an individual differences perspective, focusing on motivational variables such as "what kind of learner" [2], it is also important to approach SRL from an intra-individual perspective, such as "how each learner uses the strategy".

The authors cite Schraw and Moshman which is the underpinning research that informs the model used in this study. The reference is 1995 and yet no commentary on this research is given. Has it been cited or used extensively? What evidence beyond the original research supports its use? Are there any limitations with this research paper?

RESPONSE: Thank you for pointing this out. I have added a description of the relevant previous studies and added the literature that has influenced them to date. This literature appears to be cited in over 2,800 references as of September 2023 (by Google Scholar).

BEFORE: Line 21

… and the conditions (conditional knowledge). Murayama used this proposal to model a process to use a learning strategy hierarchically [10].

AFTER: Line 27

… and the conditions (conditional knowledge). They reviewed metacognitive theory prior to the publication of this literature and proposed two components of metacognition: not only "knowledge of cognition" as described above, but also "regulation of cognition" such as planning, monitoring, and evaluation. To date, their proposed model has been widely accepted, with metacognition proven to mediate achievement goals and facilitate mathematical modeling in mathematics education [10], and a list of metacognitive teaching practices for instructors to implement in biology education [11]. Furthermore, metacognitive knowledge has emerged as a key individual metacognitive trait in self-control studies [12]. Murayama used their knowledge of cognition to model [9] a process to use a learning strategy hierarchically [13].

10. Hidayat R, Zulnaidi H, Syed Zamri SNA. Roles of metacognition and achievement goals in mathematical modeling competency: A structural equation modeling analysis. PLoS ONE. 2018 Nov;13(11):e0206211. https://doi.org/10.1371/journal.pone.0206211

11. Stanton JD, Sebesta, AJ, Dunlosky, J. Fostering metacognition to support student learning and performance. CBE—Life Sciences Education. 2021 Apr;20(2):fe3. https://doi.org/10.1187/cbe.20-12-0289

12. Hennecke M, B ¨urgler S. Metacognition and self-control: An integrative framework. Psychol Rev. Advance online publication. https://doi.org/10.1037/rev0000406

What is the research aim or research question? This would be helpful at the end of the introduction to bring together how this research addresses a gap in the knowledge base.

RESPONSE: Thank you for your important remarks. We have clearly stated the purpose of this study and our expectations.

BEFORE: N/A

AFTER: Line 72

The purpose of this study is to illustrate the simplified results of the process leading up to the use of a learning strategy, in particular the hierarchical nature of the process due to metacognitive knowledge. If a learning strategy is used progressively, then (a) the learner must first comprehend the strategy to use it, and (b) after comprehending it, the learner determines whether or not to use it based on subjective benefit and cost. These predictions are supported when (a') the main effect of Knowledge on Used is observed and (b') the effects of Benefit and Cost on Used are observed in strategies with Knowledge.

Line 94: were referenced the Reading Strategies Questionnaire – “in” the Reading?

RESPONSE: Thank you for pointing this out. I have reflected it in the statement.

BEFORE: Line 94

This questionnaire’s categories and items were referenced the Reading Strategies Questionnaire developed by …

AFTER: Line 125

This questionnaire’s categories and items were referenced in the Reading Strategies Questionnaire developed by …

In Table 2 there are missing values, yet the procedure and participants section, mentions exclusion of participants where variables were missing?

RESPONSE: The description was unclear to the reader. Thank you for your valuable remarks.

BEFORE: N/A

AFTER: Line 101

To examine the hierarchy of metacognitive knowledge of reading strategy use in college students' descriptive essays, age was used as a criterion in this study to exclude graduate students and experienced working adults, who are likely to have more experience reading expository texts than college students. Regarding the missing data criteria, the rationale is that abandoning of a response in the middle of the survey is not random missing data, and it may cause bias in the results [11]. As described below, there were 28 reading strategies, and participants were asked to respond to four variables (Used, Knowledge, Benefit, and Cost) per item, so the maximum response per participant was 112. Of the 184 participants who were not excluded by the above criteria, 22 had at least one missing measurement (22 / 184 = 12%), and even the participant with the most missing measurements among these 22 had a missing number of 12 (12 / 112 = 11%).

The discussion section, introduces the overall results which do seem pretty obvious. It is good that this is reflected because declarative knowledge would be required to evaluate perceived benefit and cost and hence the choice to use such a strategy.

it suggested the necessity of teaching both the subject matter and 205 strategy in education

A section or discussion on the impact of the outcomes from this study on practice would be useful and help contextualise the findings further.

RESPONSE: Thank you for your advice on how to make the findings of my manuscript more generalizable. I have added the following.

BEFORE: N/A

AFTER: Line 250

Several recommendations can be made based on the study’s findings. Although the importance of teaching a knowledge regarding strategy has already been states [11], it is possible that learners may be referring to subjective benefit and/or cost in using strategies. Based on these considerations, it is suggested that there is a need for a phase in which students are not only taught the learning strategies throughout the class, but are also provided with assignments in which they use the strategies they are aiming to acquire, and are given feedback on how their scores have increased through their use.

---

## [Editor Report · Decision Letter 1]

23 Oct 2023

Knowing the learning strategy is not enough to use it: Example in reading strategies for Japanese undergraduates

PONE-D-23-13187R1

Dear Dr. Yamaguchi,

We’re pleased to inform you that your manuscript has been judged scientifically suitable for publication and will be formally accepted for publication once it meets all outstanding technical requirements.

Kind regards,

Iftikhar Ahmed Khan

Academic Editor

PLOS ONE
---

## [Editor Report · Acceptance letter]

9 Nov 2023

PONE-D-23-13187R1 

Knowing the learning strategy is not enough to use it:
Example in reading strategies for Japanese undergraduates 

Dear Dr. Yamaguchi:

I'm pleased to inform you that your manuscript has been deemed suitable for publication in PLOS ONE. Congratulations! Your manuscript is now with our production department. 

Kind regards, 

on behalf of

Dr. Iftikhar Ahmed Khan 

Academic Editor

PLOS ONE